



# The Effects of Ocean Surface Waves on Global Forecast in CFS Modeling System v2.0

Ruizi Shi[1], Fanghua Xu[1], Li Liu[1], Zheng Fan[1], Hao Yu[1], Xiang Li[2] and Yunfei Zhang[2]

[1]Ministry of Education Key Laboratory for Earth System Modeling, and Department of Earth System Science, Tsinghua University, Beijing 100084, China

[2] Key Laboratory of Marine Hazards Forecasting, National Marine Environmental Forecasting Center, Ministry of Natural Resources, Beijing, 100081, China

*Correspondence to*: Fanghua Xu (fxu@mail.tsinghua.edu.cn)

**Abstract.** It has been well known that ocean surface gravity waves have enormous effects on physical processes at the atmosphere–ocean interface. However, the effects of surface waves on global forecast in several days are less studied. To investigate this, we incorporated the WAVEWATCH III model into the Climate Forecast System Model version 2.0 (CFS2.0), with the Chinese Community Coupler version 2.0 (C-Coupler2). Two major wave-related processes, the Langmuir mixing and the sea surface momentum roughness, were considered. Extensive comparisons were performed against in-situ buoys, satellite measurements and reanalysis data, to evaluate the influence of the two processes on the forecast of sea surface temperature, mixed layer depth, significant wave height, and 10-m wind speed. A series of 7-day simulations demonstrate that the newly developed atmosphere-ocean-wave coupling system could improve the CFS global forecast. The Langmuir mixing parameterization could increase the vertical movement of water and effectively reduce the warm bias of sea surface temperature and shallow bias of mixed layer depth in the Antarctic circumpolar current in austral summer, whereas the significant wave height and 10-m wind speed are insensitive to it. On the other hand, the modified momentum roughness length could significantly reduce the overestimated 10-m wind speed and significant wave height in mid-high latitudes. This is because the enhanced frictional dissipation at high wind speed could





reduce 10-m wind speed and consequently decrease the significant wave height. But its effect on the
temperature structure in upper ocean is less obvious.
**1 Introduction**
Ocean surface gravity waves play an important role in modifying physical processes at the atmosphere–
ocean interface, which can influence momentum, heat, and moisture fluxes across the air-sea interface
(Li and Garrett 1997; Taylor and Yelland, 2001; Moon et al., 2004; Belcher et al., 2012; Moum and
Smyth, 2019). For instance, ocean surface waves can modify the ocean surface roughness to influence
the marine atmospheric boundary layer and thus change the momentum, latent heat, and sensible heat
transfer (Taylor and Yelland, 2001; Moon et al., 2004). The breaking waves inject turbulent kinetic
energy in the upper ocean, which can enhance the mixing process (Terray et al. 1996). Nonbreaking
surface waves can also affect mixing in the upper ocean by adding a wave-related Reynolds stress (Qiao
et al., 2004). The wave-related Stokes drift interacts with the Coriolis force and produces the Coriolis-
Stokes force (Hasselmann 1970). The shear of Stokes drift is a critical reason for the generation of
Langmuir circulation, which could significantly deepen the mixed layer by strong vertical mixing process
both at climate scale (Li and Garrett 1997; Belcher et al., 2012) and at weather scale (Kukulka et al.,
2009). If sea ice is present, the interaction of wave, ocean and atmosphere is further complicated (Kohout
and Meylan, 2008; Squire et al., 2009).
As Fox-Kemper et al. (2019) expected, the improvement to atmosphere-ocean coupling with a better
presentation of the effects of surface gravity waves, is one of the challenges and focuses in ocean
modeling for the next decade. Regional coupled models were developed to study tropical cyclones, storm
surge and other coastal processes at small or medium scales (e.g. Prakash et al., 2018; Ricchi et al., 2017;



Pianezze et al., 2018; Wu et al., 2019). The Coupled Ocean-Atmosphere-Wave-Sediment Transport
Modeling System (COAWST) developed by Warner et al. (2010) is one of well-known fully-coupled
models, which includes effects of wave-state-dependent ocean surface roughness, radiation stress,
bottom stress and Stokes drift-related processes. The COAWST has been well applied in various
locations such as the South China Sea (Sun et al., 2019; Wu et al., 2019), Bay of Bengal (Prakash et al.,
2018) and Mediterranean (Ricchi et al., 2017). On the other hand, the coupled models with a wave
component at global scale were primarily developed for long-term climate research (e.g. Qiao et al. 2010;
Breivik et al. 2015; Chune, et al. 2018; Fan et al., 2012; Fan and Griffies, 2014; Li et al. 2016, 2017).
The effects of waves on short term forecast at global scale have been considered negligible for long time.
Since the impact of wave-related processes is important not only for the synoptic processes but also for
the frequent interactions at multiple spatial scales as aforementioned, it is of great interest to investigate
the effects of surface ocean waves on short-term forecast in a global atmosphere-ocean-wave system
with suitable presentations of wave-related processes.
To realize a fully-coupled modeling system, establishing suitable connections between the wave
component and the atmosphere/ocean component are crucial. In coupled systems, commonly the
atmosphere and ocean components provide 10-m winds and surface currents, sometimes with other
variables such as sea surface temperature and water depth, to the wave model as forcing fields (Chen et
al. 2007; Warner et al. 2010; Breivik et al. 2015; Li et al. 2016; Pianezze et al., 2018). Compared to a
single wave model, in which the inputted reanalysis datasets usually have an interval more than 3 hours,
the forcing fields in the wave component have a finer time interval (Fan et al., 2012). Meanwhile, the
wave component sends wave parameters, such as wave length, period and significant wave height, to the
atmosphere and ocean components. These wave parameters could be used in various wave-related



parameterizations. In this study, we coupled the WAVEWATCH III to the Climate Forecast System
Model (CFS) using the Chinese Community Coupler version 2.0 (C-Coupler2). We mainly considered
two effects induced by waves at the ocean-atmosphere interface, surface roughness and Langmuir cells
induced mixing. This is because both processes have strong influences on momentum and energy fluxes
across the air-sea interface and could effectively improve the simulation results (e.g. Fan et al., 2012;
Fan and Griffies, 2014; Li et al. 2016, 2017). Four series of 7-day forecasts were produced with this
system. The performance of the system was then compared with observations and reanalysis data.
Sensitivity experiments with various wave parameterizations were carried out to evaluate the
contributions of surface roughness and Langmuir mixing to the changes of atmosphere and ocean. In
addition, the performance of various wave parameterizations was evaluated as well. The analysis is
structured as follows: methods and a set of experiments with various parameterizations are described in
Section 2; the observation and reanalysis data are introduced in Section 3, and the results of experiments
are evaluated against these available data in Section 4; a summary and discussion follow in Section 5.
**2 Methods and Experiments**
**2.1 Coupling WAVEWATCH III with CFS2.0**
The version 5.16 of WAVEWATCH III (WW3; WAVEWATCH III Development Group, 2016)
developed by the National Oceanic and Atmospheric Administration/National Centers for Environmental
Prediction (NOAA/NCEP) has been incorporated into the Climate Forecast System Model, version 2.0
(CFS2.0; Saha et al., 2014) as a new model component. The latitude range of WW3 is 78°S–78°N with
a spatial resolution of 1/3°; the frequency range is 0.04118-0.4056Hz and the total number of frequencies
is 25; the number of wave directions is 24 with a resolution of 15°; the maximum global time step is 450



s and the minimum source term time step is 300 s. The CFS contains two components, the global
forecasting system (GFS; details about the GFS are available at
http://www.emc.ncep.noaa.gov/GFS/doc.php) as the atmosphere component and the modular ocean
model version 4 (MOM4; Griffies et al., 2004) as the ocean component. The MOM4 is integrated on a
nominal 0.5° horizontal grid with enhanced horizontal resolution in the tropics, and has 40 vertical levels;
the vertical spacing is 10 m in the upper 225 m, and then increases in unequal intervals to the bottom at
4478.5 m. The GFS uses a spectral triangular truncation of 126 waves (T126) in the horizontal, which is
equivalent to a grid resolution of nearly 100 km, and 64 sigma-pressure hybrid layers in the vertical. The
time steps of both MOM4 and GFS are 180 s.
This coupled system uses the Chinese Community Coupler version 2.0 (C-Coupler2; Liu et al., 2018) for
interpolating and passing variables between its atmosphere, ocean, and wave components, to guarantee
each component receives inputs and supplies outputs on its own grid. The C-Coupler2 is a common,
flexible and user-friendly coupler, which contains dynamic 3-D coupling system and enables variables
to remain conserved after interpolation. The variables are exchanged every other time step, which in
atmosphere and ocean components is 180 s, and in wave component is 450 s.
A schematic diagram of the coupled atmosphere-ocean-wave system is shown in Fig. 1. As illustrated,
WW3 is two-way coupled with MOM4 and GFS, through the C-Coupler2. WW3 is forced by 10-m wind
from GFS and sea surface current from MOM4, and then generates and evolves the wave action density
spectrum. Meanwhile, the momentum roughness length is passed to GFS from WW3 (see section 2.3),
and the surface Stokes drift velocity is passed to MOM4 (see section 2.2). In this study, both the CFS
and WW3 use warm boots; the daily initial fields at 00:00 for CFS are generated by the real time
operational Climate Data Assimilation System (CDAS; Kalnay et al., 1996), downloaded from the CFS





official website (http://nomads.ncep.noaa.gov/pub/data/nccf/com/cfs/prod). To get initial conditions for
WW3, a single WW3 model is set up synchronously (see section 2.4).

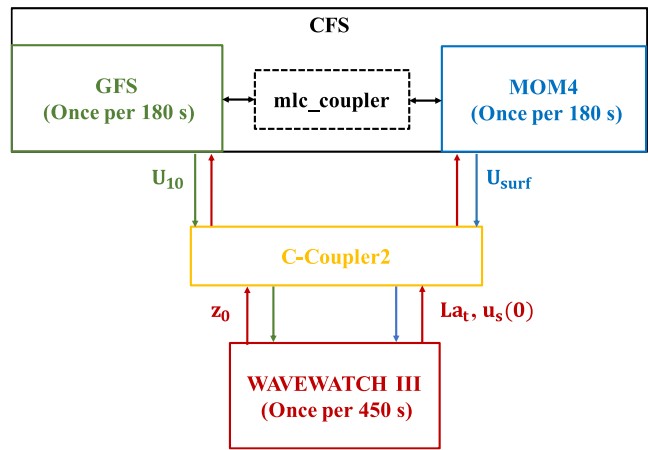


**Figure 1.** A schematic diagram of the atmosphere-ocean-wave coupled modeling system. *The arrows* indicate the
coupled variables that are passed between the model components. In the diagram, $z_0$, $La_t$, $u_s(0)$, $U_{10}$, and $U_{surf}$
are momentum roughness length, turbulent Langmuir number, surface Stokes drift velocity, 10-m wind and surface
current, respectively.
**2.2 Parameterizations of Langmuir Mixing**
**2.2.1 McWilliams and Sullivan (2000) Parameterization**
McWilliams and Sullivan (2000) improved the turbulent velocity scale W in KPP by introducing an
enhancement factor $\varepsilon$, to account for both boundary layer depth changes and nonlocal mixing by
Langmuir turbulence. In their work, they indicated W (W=$ku_*/\phi$, where $u_*$ is the surface friction
velocity, $\phi$ is the dimensionless flux profile, and $k$=0.4 is the von Kármán constant) varies in proportion
to the turbulent Langmuir number, that is,

$$W = \frac{ku_*}{\phi}\varepsilon, \qquad\qquad (1)$$

$$\varepsilon = \sqrt{1 + 0.08La_t^{-4}}, \qquad\qquad (2)$$



where $La_\mathrm{t}$ is the turbulent Langmuir number. And $La_\mathrm{t}$ is defined as

$$La_\mathrm{t} = \sqrt{\frac{u_*}{|u_s(0)|}}, \qquad (3)$$

with $u_s(0)$ is the surface Stokes drift velocity. Hereafter, we refer to this parameterization as MS2K.
Furthermore, the enhanced W will influence the calculation of boundary layer depth. In KPP the
boundary layer depth is determined as the smallest depth at which the bulk Richardson number equals
the critical value $Ri_\mathrm{cr} = 0.3$, that is,

$$Ri_b(h) = \frac{gh[\rho_r - \rho(h)]}{\rho_0[|u_r - u(h)|^2 + W^2]} = Ri_\mathrm{cr}, \qquad (4)$$

where $g$ is acceleration of gravity, $\rho$ is density, $u$ is velocity, $\rho_r$ is surface density, $u_r$ is surface
velocity, $\rho_0$ is an average value and $h$ is the boundary layer depth. Hence, when W is enhanced, the
boundary layer depth $h$ is deepened accordingly.
**2.2.2 Van et al. (2012) Parameterization**
Based on the work of McWilliams and Sullivan (2000), Van et al. (2012) proposed a different formula
for the enhancement factor, and a projected Langmuir number considering the misalignment of winds
and waves. They suggested a projected Langmuir number,

$$La_\mathrm{proj} = \sqrt{\frac{u_* \cos(\alpha)}{|u_s(0)| \cos(\theta_{ww} - \alpha)}}, \qquad (5)$$

$$\alpha \approx \tan^{-1}\left[\frac{\sin\theta_{ww}}{\frac{u_*}{u_s(0)k}\ln\left(\left|\frac{h}{Z_1}\right|\right) + \cos\theta_{ww}}\right]. \qquad (6)$$

Here $\alpha$ is the angle between wind and Langmuir cell, $\theta_{ww}$ is the angle between Stokes drift and wind,
and $Z_1$ is the four times of the significant wave height. In this case, Van et al. (2012) suggested the form
of $\varepsilon$ should be





$$\varepsilon = |cos\alpha|\sqrt{1 + (3.1La_{\mathrm{proj}})^{-2} + (5.4La_{\mathrm{proj}})^{-4}}. \qquad (7)$$

In the work of Li et al. (2016) these parameterizations corresponding to alignment and misalignment of
winds and waves (referred to as VR12-AL with $\alpha \equiv 0$ and VR12-MA with $\alpha$ not zero) were employed in
a coupled global climate model. As Li et al. (2016) illustrated, the difference between the effects of
VR12-AL and VR12-MA is not significant, owing to the limitation of coarse resolution which cannot
accurately represent the refraction by coasts and current features. Besides, the VR12-MA will certainly
increase the runtime due to increased variables to be transferred from wave to ocean. Considering all
above, we employ the VR12-AL parameterization. In VR12-AL, the $La_{\mathrm{proj}}$ (Eqn. 5) reduces to $La_{\mathrm{t}}$
(Eqn. 3).
**2.3 Parameterizations of Momentum Roughness**
In a coupled model, the estimates of momentum, latent heat, and sensible heat fluxes between atmosphere
and ocean are critically important. These fluxes are in part determined by surface roughness length, which
can be converted to drag coefficient.
In GFS, the Charnock relationship (Charnock, 1955) is used to parameterize the momentum roughness
length as

$$z_{ch} = \frac{z_0 g}{u_*^2}. \qquad (8)$$

Here $z_0$ is the roughness length, and $z_{ch} = 0.014$ is the constant Charnock number. The corresponding
scatterplot of $z_0$ in GFS versus 10-m wind speed is shown in Fig.2 (black dots). The $z_0$ in GFS
increases relatively slowly with increasing wind speed, especially at high winds.



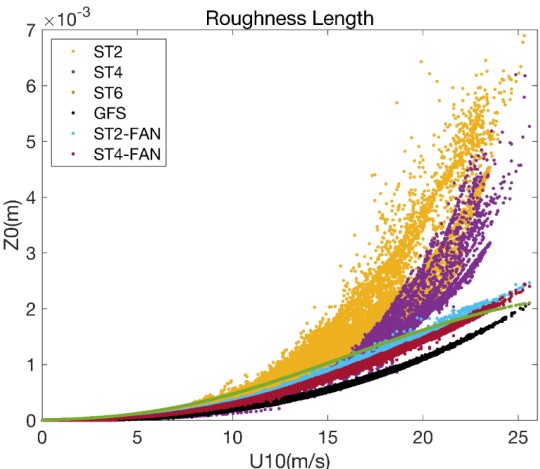


**Figure 2.** Scatterplot of momentum roughness length $z_0$ (m) in various source term packages versus 10-m wind

speed (m/s).
In WW3, input of momentum and energy by wind, and dissipation for wave-ocean interaction are two
important terms (combined as input-dissipation source term) in the energy balance equation
(WAVEWATCH III Development Group, 2016). Several different packages to calculate the input-
dissipation source term (ST) are offered in the WW3 version 5.16, and the most commonly used ones
are ST2 (Tolman and Chalikov, 1996), ST3 (Janssen, 2004; Bidlot, 2012), ST4 (Ardhuin et al., 2010),
and ST6 (Zieger et al., 2015). In ST4 package, wind speed is assumed to satisfy the traditional logarithmic
profile in the neutral boundary layer

$$U_{10} = \frac{u_*}{k} \log(\frac{z_{10}}{z_0}), \tag{9}$$

where $U_{10}$ is 10-m wind speed, and $z_{10}$ is the corresponding height (10 m). The same relationship is
also used in ST2 package (Tolman and Chalikov, 1996). And the roughness length $z_0$ is obtained from
this relationship in ST4 and ST2 (purple and yellow dots in Fig.2). Fan et al., (2012) indicated that the
$z_0$ in ST2 increases rapidly with wind speed at high winds and results in the fast-increasing drag
coefficient, which is inconsistent with the drag coefficient leveling off for extremely high wind speed in





observation and laboratory experiments. Although the rising trend of ST4 is slightly slower than that of
ST2, the rapid increase of $z_0$ at high winds still exists. For this reason, the ST6 calibrated with flux
parameterization FLX4 was proposed by Zieger et al. (2015) and accounts for the saturation of the sea
drag at high wind speeds (green dots in Fig.2). However, the drag coefficient in ST6 is calculated by
wind speed only without wave state considered, and this does not accord with the fact.
To solve these problems, Fan et al., (2012) suggested an improved parameterization for $z_0$ in ST2
(referred to as ST2-FAN). In ST2-FAN, $z_0$ is calculated by the improved Charnock relationship, in
which the Charnock number is not a constant but depends on the wave state (Moon et al., 2004), that is,

$$\frac{z_0 g}{u_*^2} = a\left(\frac{c_{pi}}{u_*}\right)^b, \qquad (10)$$

$$a = \frac{0.023}{1.0568^{U_{10}}}, b = 0.012 U_{10}, \qquad (11)$$

where $c_{pi}$ is the phase speed of dominant wind-forced waves.
In this study, we chose the ST4 package to calculate the input and dissipation term, since ST4 has shown
the best performance in the simulation of significant wave height (SWH) at global scale (compared in
section 2.4), which is consistent with the study of Stopa et al (2016). Fan et al. (2012)'s parameterization
was then applied in ST4 (referred to as ST4-FAN) to obtain new $z_0$. The estimates of $z_0$ from ST2-
FAN and ST4-FAN are shown in blue and dark red dots of Fig.2, respectively. The fast-rising trend of
$z_0$ at high wind speed is obviously restrained. And $z_0$ from ST4-FAN is generally smaller than that from
ST2-FAN.
**2.4 Initialization of WAVEWATCH III**
The initial wave fields were generated from the 10-day simulations starting from rest in a single WW3
model. To minimize the biases of the initial wave fields, we ran simulations with ST2, ST3, ST4, and



ST6 source terms respectively, and compared the results. Besides, two 10-m wind datasets were used
and compared as the wave model forcing, namely the Cross-Calibrated Multi-Platform (CCMP; Atlas et
al., 2011) data and the fifth generation European Centre for Medium-Range Weather Forecasts (ECMWF)
Reanalysis (ERA5; Hersbach and Dee, 2016) data. After comparisons, the ST4 source term with ERA5
wind forcing, which generated the minimum SWH bias (Table S1 in the supplementary), was applied to
generate initial wave fields for all experiments listed in Table 1.
**Table 1**. List of Numerical Experiments

| Experiments | WW3 to MOM4 | WW3 to GFS |
|---|---|---|
| CTRL | None | None |
| VR12-AL-ONLY | VR12-AL parameterization | None |
| Z0-ONLY | None | $z_0$ from ST4 |
| VR12-AL-Z0 | VR12-AL parameterization | $z_0$ from ST4 |
| MS2K-Z0 | MS2K parameterization | $z_0$ from ST4 |
| VR12-AL-Z0-FAN | VR12-AL parameterization | $z_0$ from ST4-FAN |

**2.5 Set of Experiments**
A series of numerical experiments was conducted to evaluate the effects of Langmuir mixing
parameterizations and momentum roughness lengths on the ocean and atmosphere in four 7-day periods,
January 3 to 10, 2017, July 1 to 8, 2018, August 3 to 10, 2018, and January 1 to 8, 2019.
The reference experiment (CTRL) is a one-way coupled experiment, in which GFS and MOM4 provide
10-m wind and sea surface current to WW3, whereas no variables transmission from WW3 to CFS. The
results of CFS in CTRL are consistent with the corresponding CFS Reanalysis data (Saha et al., 2010).
For each time period, five sensitivity experiments were carried out. The first is VR12-AL-ONLY
simulation, in which the VR12-AL parameterization is added in MOM4. The second is Z0-ONLY



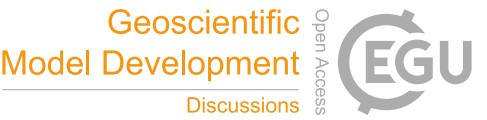

simulation, in which the original $z_0$ in GFS is replaced by $z_0$ from WW3 ST4 source term. Then based
on the VR12-AL-ONLY simulation, the VR12-AL-Z0 simulation is performed with $z_0$ from WW3 ST4
source term in GFS. The MS2K-Z0 simulation is similar to the VR12-AL-Z0 simulation, but using the
MS2K instead of VR12-AL. The last experiment is the VR12-AL-Z0-FAN, in which $z_0$ is generated
by the ST4-FAN source term in WW3 and other settings remain the same as VR12-AL-Z0.
**3 Data**
Sea surface temperature (SST), profiling temperature and salinity, 10-m wind speed (WSP10), and
significant wave height (SWH) from observations and reanalysis datasets are used to evaluate the
simulation results.
The daily average satellite Optimum Interpolation SST (OISST) data is obtained from the National
Oceanic and Atmospheric Administration (NOAA), with 0.25°×0.25° resolution (Reynolds et al., 2007;
https://www.ncdc.noaa.gov/oisst). The global Argo observational profiles of temperature and salinity (Li
et al., 2019) is from China Argo Real-time Data Center (www.argo.org.cn). The fifth generation
European Centre for Medium-Range Weather Forecasts (ECMWF) Reanalysis (ERA5) datasets of
WSP10 and SWH with a spatial resolution of 0.5° and a temporal resolution of 1 hour are also used
(Hersbach and Dee, 2016; https://cds.climate.copernicus.eu/cdsapp#!/dataset/_reanalysis-era5-single-
levels). Additionally, the WSP10 and SWH observations from the available National Data Buoy Center
(NDBC) buoy data (https://www.ndbc.noaa.gov), and the along-track SWH from Jason-3 satellite
measurements (https://aviso-data-center.cnes.fr) Geophysical Data Record (GDR) with precise orbit and
an orbital velocity of 7.2 km/s are applied for comparison purposes.



## 4 Results

### 4.1 SST and Mixed Layer Depth (MLD)

The application of Langmuir mixing parameterization in KPP can change the SST, because the modified

turbulent vertical velocity scale enhances surface ocean mixing, which tends to reduce SST. In the study,

the distribution pattern of biases is almost unchanged within 7 days. But the magnitude of the biases

slightly increases with time, and the influences of parameterizations also become more obvious. Without

loss of generality, we compared the distributions of SST on the 4th day as the intermediate state, and the

similar distributions on the last day are also shown in Figs. S1&S2 of the supplementary. Figure 3 and

Figure 4 show the distribution maps of daily average SST in CTRL (Fig.3a&4a), its bias (Fig.3b&4b)

and percentage absolute difference of the bias from experiments versus the CTRL (Fig. 3c-g and Fig. 4c-

g), on January 7, 2017 and August 7, 2018 (the 96th-120th hours), respectively. Here the bias is defined

as SST in CTRL minus OISST. And to highlight the differences of other experiments versus the CTRL,

the percentage absolute differences (PAD) of the bias are computed as $PAD = \frac{|\hat{y}_s - y| - |\hat{y}_c - y|}{|y|} \times 100\%$,

where $y$ is OISST, $\hat{y}_c$ is simulated SST in CTRL and $\hat{y}_s$ is simulated SST in other experiments (Fig.

3c-g and Fig. 4c-g). A negative value of PAD indicates that the error is smaller compared to CTRL, and

vice versa.



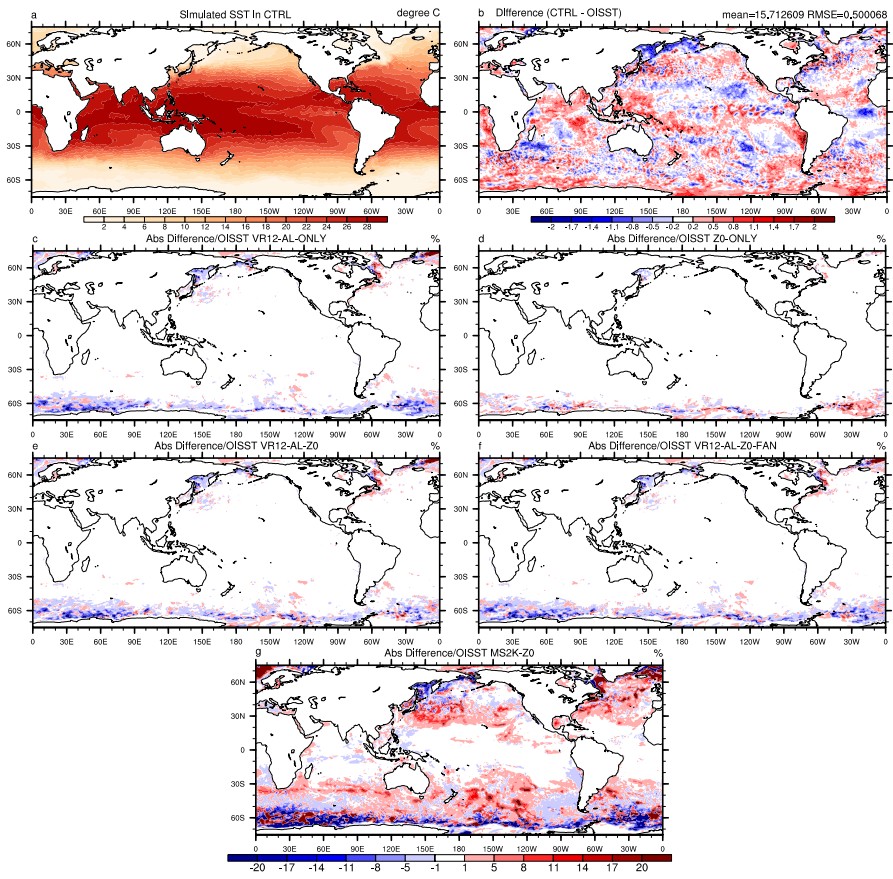

**Figure 3.** The daily average SST (℃) in CTRL, its bias in CTRL and percentage absolute difference of bias on

January 7, 2017: **a** the SST in CTRL, **b** the SST bias between CTRL and OISST (CTRL minus OISST), **c/d/e/f/g** the

percentage absolute difference between VR12-AL-ONLY/Z0-ONLY/VR12-AL-Z0/VR12-AL-Z0-FAN/MS2K-Z0

and CTRL. The absolute difference is a percentage computed as $\mathrm{PAD} = \frac{|\widehat{y_s} - y| - |\widehat{y_c} - y|}{|y|} \times 100\%$, where $y$ is OISST,

$\widehat{y_c}$ is simulated SST in CTRL and $\widehat{y_s}$ is simulated SST in other experiments, so a negative value means that the

error is smaller than that of CTRL, and vice versa.



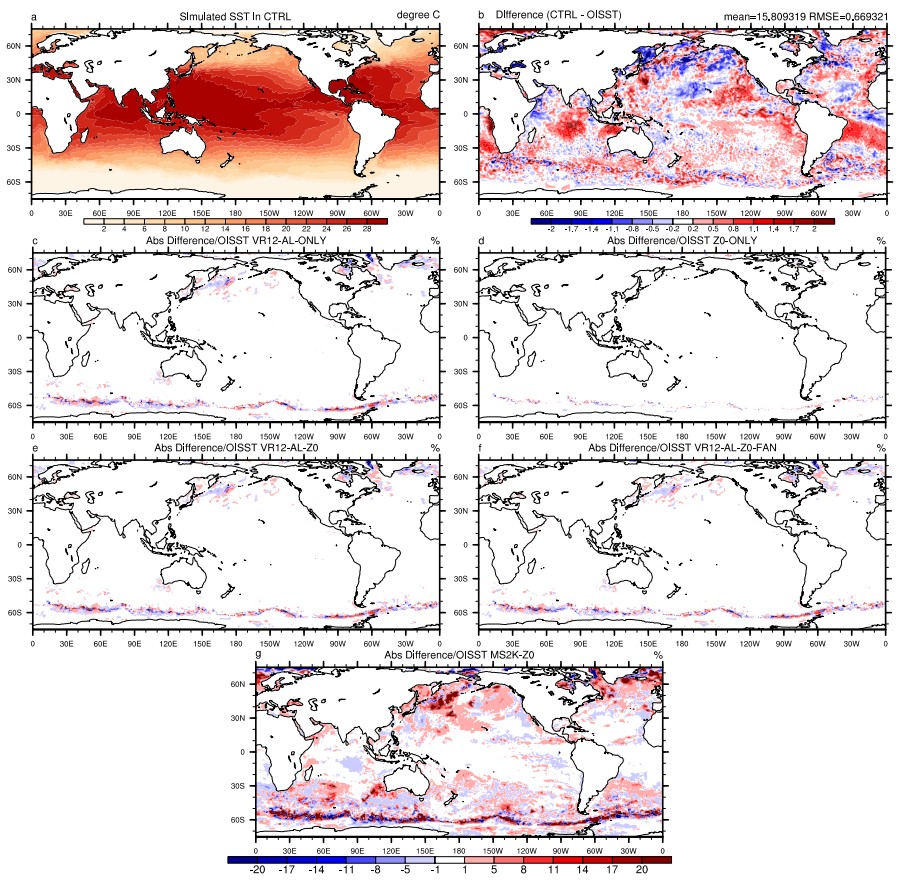

249

**Figure 4.** As Fig. 3, but for the daily average SST (℃) in CTRL, its bias in CTRL and absolute difference of bias on August 7, 2018.

As shown in Fig. 3a and 3b, the global mean SST in CTRL is approximately 15.71℃, which is close to

the mean SST from OISST (approximately 15.22℃), and the average RMSE is about 0.50℃ in CTRL.

The simulated SST is generally overestimated, and the large biases (>1.10℃) are mainly distributed at

locations with active mesoscale vortices and frontal instability, such as the Kuroshio extension, the

Peruvian upwelling, the Gulf Stream and the Antarctic Circumpolar Current (ACC; Fig. 3b). The

Langmuir mixing can enhance the vertical turbulent velocity, and thus after introducing it the simulations

should cool the surface waters and reduce warm biases in locations where Stokes drift related turbulence





kinetic energy is large (Belcher et al., 2012; Li et al. 2016). The warm bias of SST in VR12-AL-ONLY
(Fig. 3c) is clearly decreased near the ACC, because of the strong Langmuir mixing. More than 5% bias
reduction is achieved. The SST estimates are also improved by 5-11% in the Okhotsk Sea and the Bering
Sea. But there is no clear change elsewhere. This is consistent with the distribution of relatively high
SWH (Fig. 6a). In Fig. 3d, the SST improvements vanish, whereas in Fig. 3e and 3f the bias distributions
are almost identical to Fig. 3c, indicating that the SST is insensitive to the change of surface roughness
($z_0$). In contrast, the biases from MS2K-Z0 (Fig. 3g) get worse in general, due to too much mixing
induced by MS2K parameterization, which has cooled down the surface ocean greatly. As a result,
although the warm bias in ACC is greatly reduced, the cold bias is enhanced in mid latitudes (Fig. 3g).
Compared to the simulations in January, the simulations with VR12-AL parameterization in August show
less improvements, especially in ACC, where the SST bias even partially increases (Fig. 4c, e, f) because
of the reduced warm bias during austral winter in the south of 50°S (Fig. 4b). These results are consistent
with the studies of Belcher et al. (2012) and Li et al. (2016), which indicated that the improvements of
simulation by Langmuir mixing parameterizations in Southern Ocean are obvious mainly in austral
summer but not winter. To examine the robustness of these variations, two tests from January 1 to 8,
2019 and from July 1 to 8, 2018 were conducted. The results are in good agreement with the previous
simulations (Figs. S7&S8 in the supplementary). Similarly, the SST in August also becomes too cold in
MS2K-Z0 (Fig. 4g), especially in the mid latitude regions of the Northern Hemisphere, where the SST
is already underestimated in CTRL (Fig. 4b).
In order to further evaluate the direct effect of Langmuir mixing parameterizations, we compared the
mixed layer depth (MLD) of all experiments with that of Argo profiles, since MLD could be deepened
by the Langmuir mixing. The simulated temperature and salinity were interpolated onto the positions of





Argo profiles at the nearest time. The MLD was then estimated as the depth where the change of potential
density reaches the value corresponding to a 0.2°C decrease of potential temperature with unchanged
salinity from surface (de Boyer Montégut et al., 2004). Considering the less improvements to the SST
simulation by Langmuir mixing in August than in January, here we only compared the results of MLD
in January, 2017. Comparisons of the MLDs between numerical experiments and Argo data in the ACC
(0-360°E, 45-78°S) are shown in Fig. 5. Both the MS2K (dark red) and VR12-AL (yellow, blue and
green) parameterizations lead to deepened MLD, compared to CTRL (orange). Noticeably, too much
mixing introduced by MS2K parameterization results in the over-deepening of MLD. Considered the
enhanced mixing effect resulted by Langmuir turbulence, when the simulated MLD in CTRL is shallower
than observation (black), the bias is reduced in VR12-AL, such as the period from 6:00 on January 9 to
0:00 on January 10. However, when the simulated MLD in CTRL is overestimated, application of VR12-
AL parameterization tends to increase the bias on the contrary, such as 0:00 on January 9. All in all, the
biases of MLDs are reduced by Langmuir mixing. In addition, similar to the SST simulations, the
differences of MLDs generated by different $z_0$ in VR12-AL-ONLY (yellow), VR12-AL-Z0 (blue) and
VR12-AL-Z0-FAN (green) are quite few, indicating that the effect of surface roughness on upper ocean
is not significant. This is also consistent with the fact that the result of Z0-ONLY (purple) has little
difference with CTRL.



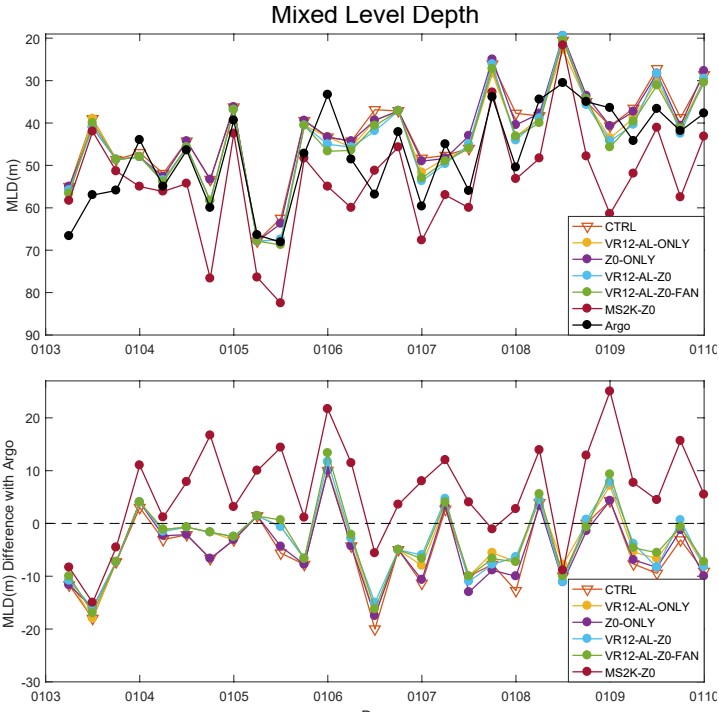


**Figure 5.** The time series of domain-averaged (0-360°E, 45-78°S) mixed layer depth (MLD; m; upper panel) and
MLD bias versus Argo profile data (simulations minus Argo; lower panel). The time intervals are 6 hours.
**4.2 Significant Wave Height (SWH) and Wind Speed at 10 m (WSP10)**
To evaluate the performance of the wave simulation, the simulated SWHs were compared with Jason-3
GDR along-track quality-checked altimeter measurements and the ERA5 reanalysis data. Here we
compared the Jason-3 data and simulations at 00:00 (the Jason-3 data within 20 min from 00:00 were
applied), and interpolated the simulated SWHs at 00:00 onto the satellite orbit. The 7-day averaged SWH
correlation coefficients, skill scores and RMSEs from 00:00 on Jan 3, 2017 to 00:00 on Jan 10, 2017 are
documented in Table 2. Compared with CTRL, experiments with coupled $z_0$ show improvements to the
SWH simulation, especially the VR12-AL-Z0-FAN. Although the improvements are not quite large,
because those are the global average results, whereas the significant improvements mainly distribute in





the mid-high latitudes (Fig.6&7). The best results (with the highest correlation coefficient, the lowest
RMSE and the highest skill score) in 6 experiments are marked in bold, all of which are from VR12-AL-
Z0-FAN. Compared with CTRL, the RMSE in VR12-AL-Z0-FAN reduces 5.0%. In addition, we also
calculated the difference between Jason-3 data and ERA5 reanalysis data (Table 2). The small bias
suggests that the ERA5 data is reliable for comparison with the global distribution of simulations. And
remarkably, the results of VR12-AL-Z0-FAN are close to ERA5. In the simulation starting from Aug 3,
2018, the correlation coefficient, skill score and RMSE of CTRL are 0.79, 0.86 and 0.68, and those of
ERA5 are 0.88, 0.93 and 0.43. The difference between CTRL and ERA5 is larger, so there is more
potential for improvement. Similarly, the best results in 6 experiments are still from VR12-AL-Z0-FAN,
of which the RMSE is 0.61 and reduces 10.0%.
**Table 2**. 7-day Averaged Correlation Coefficient, RMSE and Skill Score of SWH in Simulations and ERA5 versus
the Jason-3 Observation from 00:00 on Jan 3, 2017 to 00:00 on Jan 10, 2017. *Bold marks* represent the highest
correlation coefficient, the lowest RMSE and the highest skill score (except ERA5); the RMSE and skill score (SS)
are calculated as RMSE=$\sqrt{\sum_{i=1}^{n}(\hat{y}_i - y_i)^2/n}$ and SS=$1 - \frac{\sum_{i=1}^{n}(\hat{y}_i - y_i)^2}{\sum_{i=1}^{n}(|\hat{y}_i - \bar{y}_i| + |y_i - \bar{y}_i|)^2}$, respectively, where $\hat{y}_i$ is simulated
value or ERA5 data, $y_i$ is Jason-3 observation and $\bar{y}_i$ is the average, i=1,n and n is the total number of
measurements in the Jason-3 orbit.

| | Correlation Coefficient (P<0.01) | RMSE | Skill Score |
|---|---|---|---|
| CTRL | 0.85 | 0.60 | 0.91 |
| VR12-AL-ONLY | 0.85 | 0.61 | 0.90 |
| Z0-ONLY | 0.85 | 0.58 | 0.91 |
| VR12-AL-Z0 | 0.85 | 0.58 | 0.91 |
| MS2K-Z0 | 0.82 | 0.60 | 0.89 |
| VR12-AL-Z0-FAN | **0.86** | **0.57** | **0.92** |
| ERA5 | 0.87 | 0.51 | 0.92 |

To further investigate the effect of wave-related processes on the simulated distribution of SWH biases,





we also compared the simulated SWH with the ERA5 data. Figure 6 and Figure 7 show the distributions
of SWHs in CTRL (Fig.6a&7a), its bias (Fig.6b&7b) and percentage absolute difference of bias from
experiments versus the CTRL (Fig. 6c-g and Fig. 7c-g) at 00:00 on January 7, 2017 and August 7, 2018
(the 96[th] hour), respectively. On January 7, 2017, the global mean SWH in CTRL is approximately 2.50
m, which is higher than the mean SWH from ERA5 (approximately 2.31 m). The average RMSE is about
0.48 m in CTRL. Large biases (> 1.0 m) appear in the ACC area and the mid-high latitudes of the
Northern Hemisphere (Fig.6a&b). On August 7, 2018, the global mean SWH in CTRL is approximately
2.65 m and higher than that from ERA5 (approximately 2.35 m) with 0.60 m RMSE. The high SWH
areas in the mid-high latitudes of the Northern Hemisphere during January disappeared (Fig. 7a) with
reduced overestimated bias (Fig.7b), whereas the SWHs in ACC became higher with the maximum bias
of more than 3 m. In VR12-AL-ONLY experiments (Fig.6c&7c), compared with the CTRL, there are
few differences, which indicates that the introducing of Langmuir mixing has little influence on wave
state. Noticeably, in Z0-ONLY and VR12-AL-Z0 after introducing the wave-related $z_0$ to GFS, the
overestimated biases have decreased, and in most regions the improvements are more than 5% (Fig.6d&e;
Fig.7d&e). Compared Fig. 6b (7b) and Fig. 6d (7d), it is clear that the improvements mainly appear in
regions where SWHs are overestimated, indicating that the wave-related $z_0$ can reduce the SWH. In
VR12-AL-Z0-FAN (Fig.6f&7f), the $z_0$ from ST4-FAN parameterization (Fig. 2) has resulted in the
global reduction of SWHs. As a result, the bias has decreased (increased) in areas where SWHs are
overestimated (underestimated) in CTRL (Fig.6f&7f). The results in MS2K-Z0 are similar with that in
VR12-AL-Z0 (Fig.6g&7g), and this again illustrates that the SWH simulation is not as sensitive to
Langmuir mixing as to $z_0$.





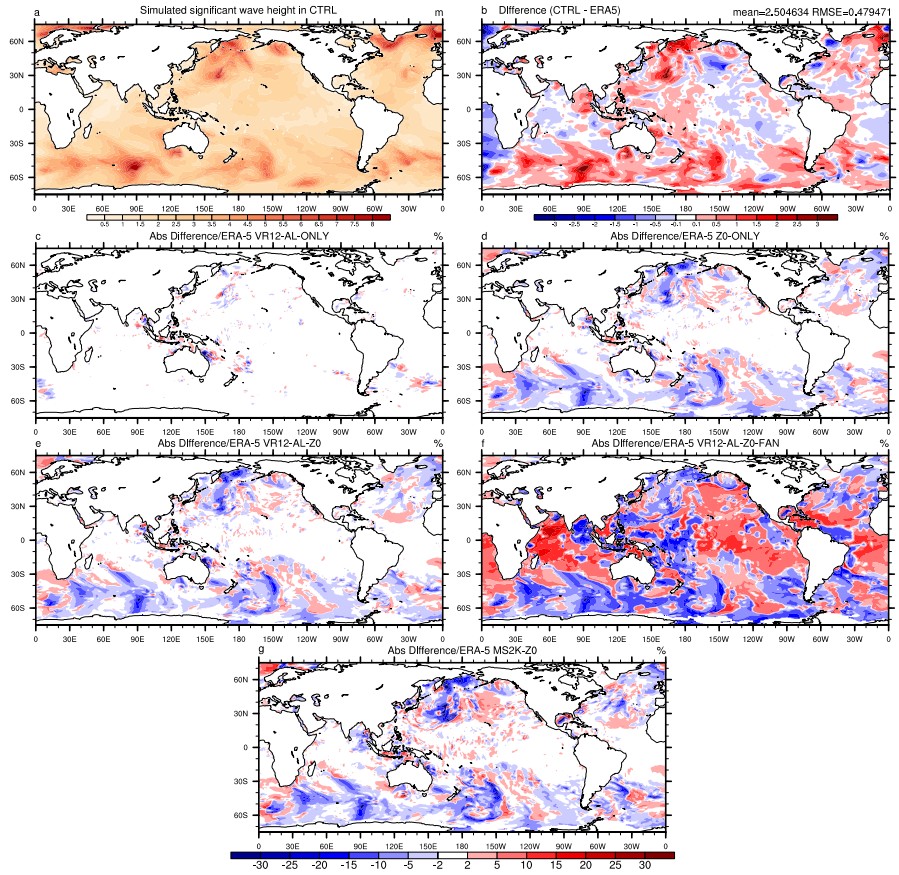

**Figure 6.** The SWH (m) in CTRL, its bias in CTRL and percentage absolute difference of bias on January 7, 2017:

**a** the SWH in CTRL, **b** the SWH bias between CTRL and ERA5 (CTRL minus ERA5), **c/d/e/f/g** the percentage

absolute difference between VR12-AL-ONLY/Z0-ONLY/VR12-AL-Z0/VR12-AL-Z0-FAN/MS2K-Z0 and CTRL.

The absolute difference is a percentage computed as $\text{PAD} = \frac{|\hat{y}_s - y| - |\hat{y}_c - y|}{|y|} \times 100\%$, where $y$ is the SWH from ERA5,

$\hat{y}_c$ is simulated SWH in CTRL and $\hat{y}_s$ is simulated SWH in other experiments, so a negative value means that the

error is smaller than that of CTRL, and vice versa.

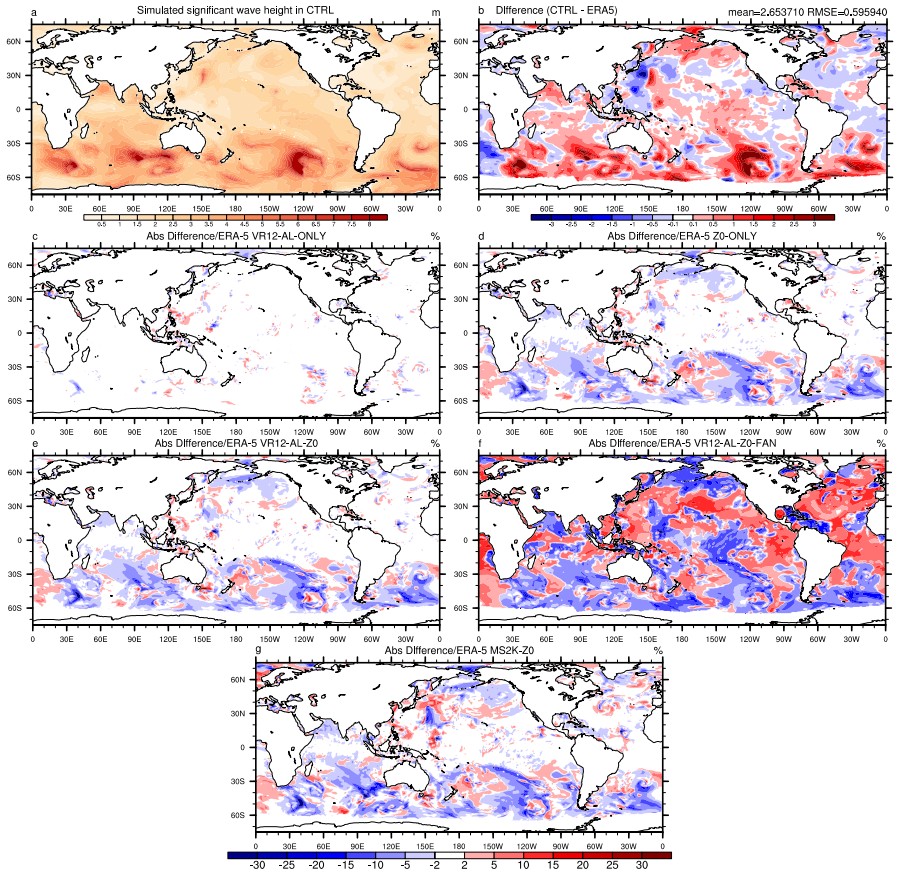

**Figure 7.** As Fig. 6, but for the SWH (m) in CTRL, its bias in CTRL and absolute difference of bias on August 7, 2018.

The waves in WW3 are mainly generated by the 10-m winds, since the effect of sea surface current is much weaker. The comparisons of the WSP10 from numerical experiments with the ERA5 wind data (Fig.8&9) indicate that the overestimated WSP10 (red shaded areas in Figs.8b&9b) could lead to the overestimated SWH (red shaded areas in Fig.6b&7b). In Z0-ONLY and VR12-AL-Z0 (Fig.8d&e; Fig.9d&e), after introduced the $z_0$ from ST4 in GFS, the biases of overestimated WSP10 are reduced and so are the biases of the SWH. In VR12-AL-Z0-FAN, the decrease of WSP10 is slightly weaker than those in VR12-AL-Z0 (Figs. 8f&9f), due to the $z_0$ from ST4-FAN which is lower than that from ST4



365 at high winds (Fig. 2). With the combined effect of strong mixing and surface roughness in MS2K-Z0,

366 the WSP10s decrease more (Figs.8g&9g). The similar SWH and WSP10 distributions at the last second

367 (the 168th hour) are also shown in Figs. S3-S6 of the supplementary, with both the increase and decrease

368 in biases due to the parameterizations becoming stronger. Moreover, the additional tests from January 1

369 to 8, 2019 and from July 1 to 8, 2018 also demonstrate the robustness of these results (Figs. S9-S12 in

370 the supplementary).

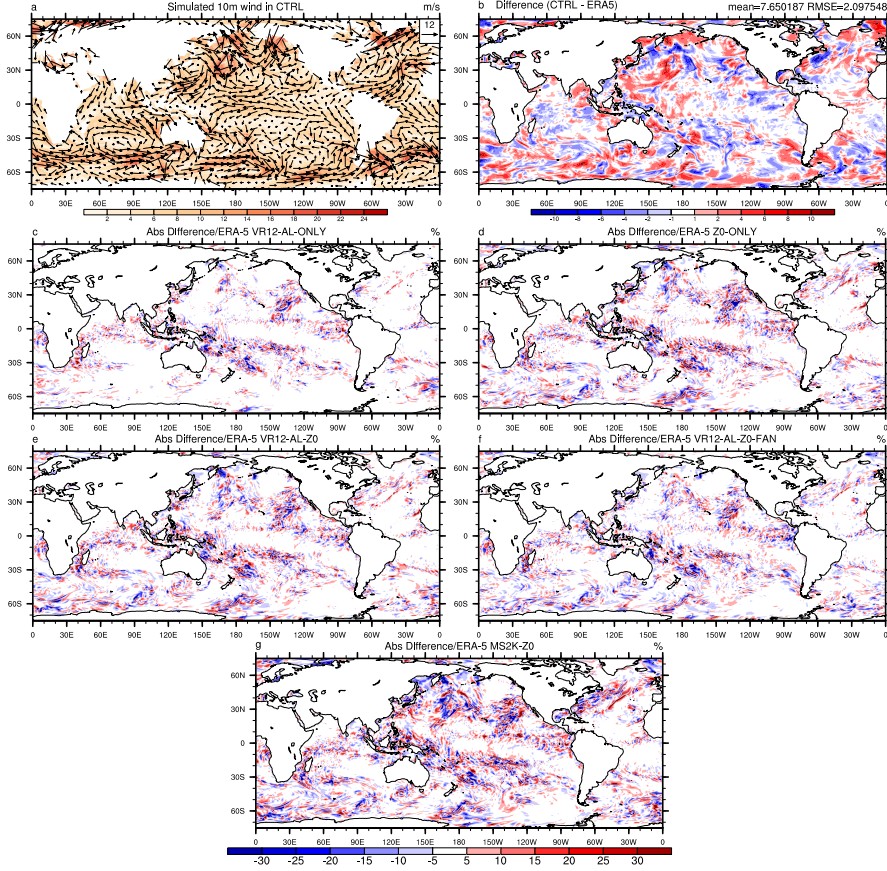

371

**Figure 8.** The WSP10 (m/s) in CTRL, its bias in CTRL and percentage absolute difference of bias on January 7,

373 2017: **a** the 10-m wind in CTRL, **b** the 10-m wind bias between CTRL and ERA5 (CTRL minus ERA5), **c/d/e/f/g**

374 the percentage absolute difference between VR12-AL-ONLY/Z0-ONLY/VR12-AL-Z0/VR12-AL-Z0-FAN/MS2K-

375 Z0 and CTRL. The absolute difference is a percentage computed as $\mathrm{PAD} = \frac{|\hat{y}_s - y| - |\hat{y}_c - y|}{|y|} \times 100\%$, where $y$ is



WSP10 from ERA5, $\hat{y}_c$ is simulated WSP10 in CTRL and $\hat{y}_s$ is simulated WSP10 in other experiments, so a
negative value means that the error is smaller than that of CTRL, and vice versa.

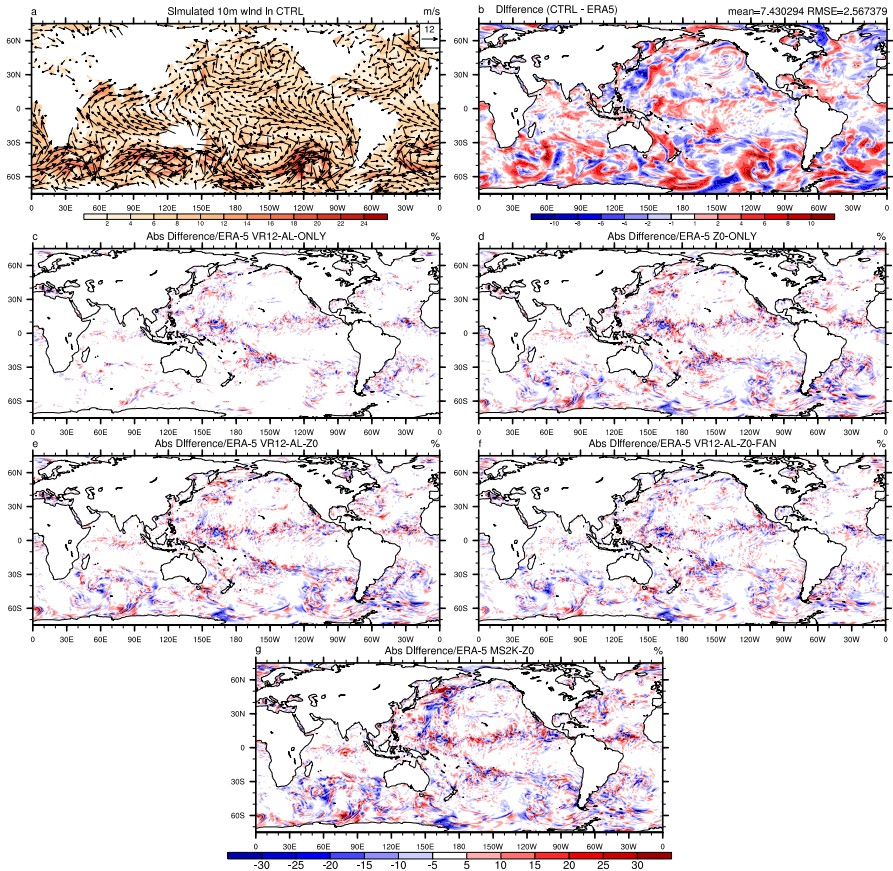


**Figure 9.** As Fig. 8, but for the WSP10 (m/s) in CTRL, its bias in CTRL and absolute difference of bias on August 7, 2018.

To demonstrate the relationship of SWH and WSP10 more clearly, we calculated the 7-day mean absolute
percentage error (MAPE) for SWH and WSP10 between simulation results and NDBC buoy data
(locations shown in Fig. 10). In general, the difference of SWH corresponds well to the difference of
WSP between CTRL and ERA5 (shaded areas in Fig. 10). The lower the MAPE, the better the
performance of the simulation. The corresponding MAPE differences compared with CTRL for the other



5 simulations are shown in Fig. 11, where a negative value means that the error is reduced versus CTRL
and vice versa. From Fig. 11, it is clear that the distribution of MAPE for SWH (Fig. 11a&c) is in
accordance with that for WSP10 (Fig. 11b&d). In the areas with overestimated SWH and WSP10, such
as location 5 on January, 2017 (Fig. 10a&b) and location 3 on August, 2018 (Fig. 10c&d), after applying
the wave-related $z_0$ in VR12-AL-Z0 and VR12-AL-Z0-FAN, the improvements of MAPEs are
manifest for both SWH and WSP10 (Fig. 11). However, for the areas with underestimated SWH and
WSP10, or with few biases, such as location 7, 8 on January, 2017 (Fig. 10a&b), the introduction of the
wave-related $z_0$ slightly increases the MAPEs (Fig. 11a&b). Besides, although it has been indicated
that the SWH is not sensitive to Langmuir mixing (Fig. 6c&7c), from Fig. 11 it is seen that as for the
WSP10, VR12-AL-Z0 could perform better than Z0-ONLY, such as at location 10 on August, 2018 (Fig.
11d). This is probably because the enhanced turbulence kinetic energy in Langmuir mixing
parameterization leads to more kinetic energy input from air to sea, which consequently results in the
reduced surface wind speed.

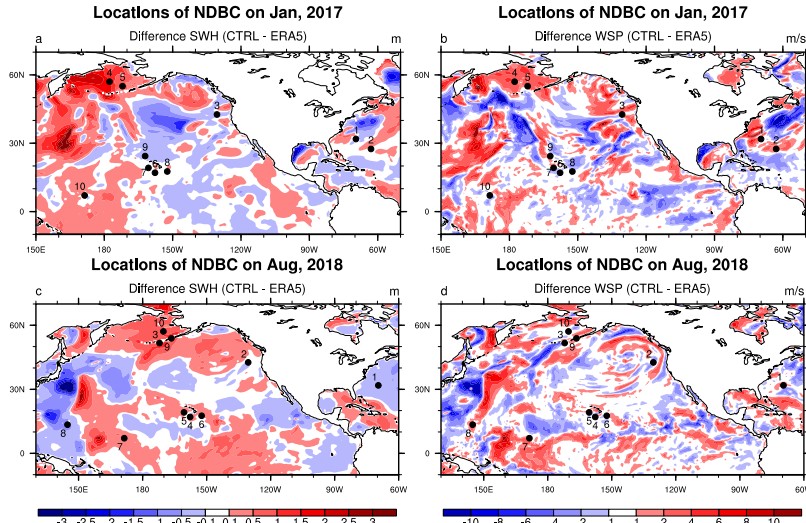


**Figure 10.** The locations of NDBC buoy data on Jan, 2017 (**a, b**) and Aug, 2018 (**c, d**); *Shaded* areas are SWH biases





(**a, c**) and WSP10 biases (**b, d**) between CTRL and ERA5 (CTRL minus ERA5).

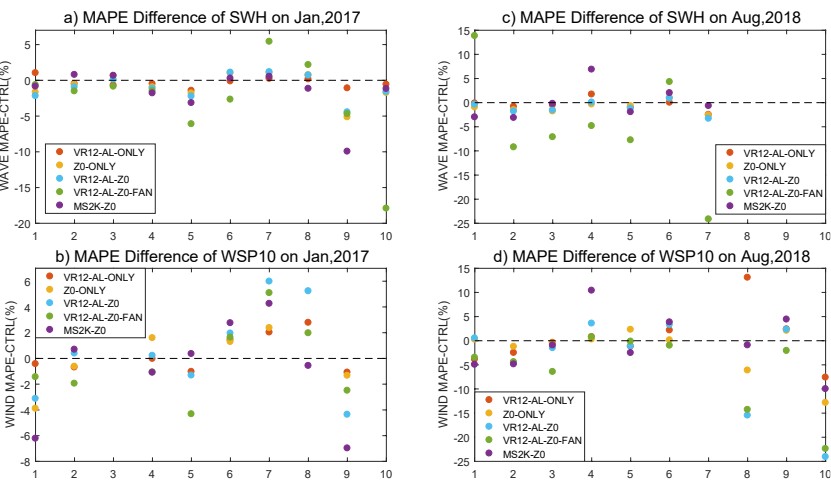


**Figure 11.** The mean absolute percentage error (MAPE) differences for SWH (**a, c**) and WSP10 (**b, d**) between
VR12-AL-ONLY/Z0-ONLY/VR12-AL-Z0/VR12-AL-Z0-FAN/MS2K-Z0 and CTRL (MAPE in VR12-AL-
ONLY/Z0-ONLY/VR12-AL-Z0/VR12-AL-Z0-FAN/MS2K-Z0 minus MAPE in CTRL) on Jan, 2017 (**a, b**) and Aug,
2018 (**c, d**); the MAPE is calculated as MAPE=$(100\%/n)\sum_{i=1}^{n}\left|\frac{\hat{y}_i - y_i}{y_i}\right|$, where $\hat{y}_i$ is simulated value, $y_i$ is NDBC
buoy observation, i=1, 7; blank is missing value.
To understand the physical mechanism of how $z_0$ parameterization affect the SWH and WSP10, the
scatterplot of various $z_0$ in ST4, ST4-FAN and original GFS need to be recalled (Fig. 2). The $z_0$ in
WW3 with ST4 source term (purple dots) is larger than the original $z_0$ in GFS (black dots) at high wind
speed (> 15m/s). The larger $z_0$ enhances frictional dissipation, therefore reduces the WSP10. And thus,
the overestimated high WSP10s in CTRL are reduced in Z0-ONLY and VR12-AL-Z0. Furthermore, in
regions like ACC, the overestimated WSP10s usually generate overestimated SWHs, therefore the
reduced WSP10s could lead to the decrease of SWHs, and then improve the SWHs simulation in Z0-
ONLY and VR12-AL-Z0. The ST4-FAN $z_0$ parameterization (dark red dots in Fig. 2) in VR12-AL-Z0-
FAN has smaller $z_0$ at high wind speed than in ST4, however the generated $z_0$ at high wind (the
threshold is less than ST4 and about 12m/s) is still larger than the original $z_0$ in GFS. Therefore, in





VR12-AL-Z0-FAN the reduction of the overestimated WSP10 in high wind areas is slightly weaker than
that in VR12-AL-Z0, and so is the overestimated SWHs. Noticeably, the $z_0$ generated by ST4-FAN is
larger than that generated by ST4 for WSP10 less than 15m/s, and enhances frictional dissipation, which
could result in decreased SWHs at low winds. Thus, in VR12-AL-Z0-FAN the SWHs decrease globally,
leading to reduced biases for overestimated SWHs but enhanced biases for underestimated SWHs.
**5 Summary and Discussion**
To investigate the role played by ocean surface gravity waves on atmosphere and ocean interface in a
coupled global atmosphere-ocean-wave modeling system in a relatively short time range, we
implemented the version 5.16 of WW3 to CFS2.0 for global oceans from 78ºS-78ºN, using the C-
Coupler2. In this coupled system, the WW3 was forced by 10-m wind generated in GFS and sea surface
current generated in MOM4. Langmuir mixing parameterizations, and momentum roughness length ($z_0$)
parameterizations were applied and compared against in-situ buoys, satellite measurements and ERA5.
The effects of waves on forecasting were examined in two winters and two summers. The results for the
same season are consistent.
The following key results were found:

1.   Langmuir mixing parameterizations could effectively reduce the SST and deepen MLD by

generating strong vertical movements for 7-day forecasting. It is beneficial for areas with large

errors of SST, such as the ACC. Particularly, the application of VR12-AL parameterization

(Van et al. 2012) could significantly reduce the warm bias of SST and shallow bias of MLD in

ACC in January, whereas its effects are nil in August. In contrast, the vertical mixing generated

by MS2K parameterization is so strong that the SST is too cold and the MLD is too deep





compared with the observations.

2.  With the application of ST4-FAN (Fan et al. 2012) for surface roughness length ($z_0$), $z_0$

becomes larger at high wind conditions, and leads to increased frictional dissipation at ocean-

atmosphere interface. As a result, the overestimated wind speeds (usually in mid-high latitudes)

are reduced. The reduced wind speeds subsequently decrease SWHs, and thus the

overestimated SWHs produced by previously overestimated wind are also reduced.

As shown in Fig. 3, the Langmuir mixing induced SST improvements are mainly distributed in mid-high
latitudes. SST biases also appeared in tropical oceans. In the work of Chune et al. (2018), the Nucleus
for European Modelling of the Ocean (NEMO) model was one-way coupled with the Météo-France wave
model (MFWAM) to refine the momentum as well as the energy flux across the air-sea interface.
Consequently, the SST cold bias in the tropics is reduced. This offers a next direction to improve the
global ocean forecast. Besides, some other processes such as nonbreaking wave-induced upper ocean
mixing (Qiao et al., 2004), may also lead to improvements.
There still remain some biases in the coupled system, probably owing to the inaccuracy of coarse
resolution, the incompleteness of direct wave-current interaction processes, and the deficiency of a
unified assimilation system. In addition, to further improve the model and eliminate the biases, as Breivik
et al. (2015) proposed, extra adjusting of the individual model components in the coupled systems is also
necessary. All of these require further efforts to investigate efficient methods that can improve the ability
of the fully coupled system.
**Code and data availability**
The code developed for the coupled system can be found under https://doi.org/10.5281/zenodo.4125726
(Shi et al., 2020), including the coupling, preprocessing, run control and postprocessing scripts. The



initial fields for CFS are generated by the real time operational Climate Data Assimilation System,
downloaded from the CFS official website (http://nomads.ncep.noaa.gov/pub/data/nccf/com/cfs/prod).
The daily average satellite Optimum Interpolation SST (OISST) data are obtained from NOAA
(https://www.ncdc.noaa.gov/oisst), and the National Data Buoy Center (NDBC) buoy data are also
obtained from NOAA (https://www.ndbc.noaa.gov). The Argo observational profiles of temperature and
salinity are available at China Argo Real-time Data Center (www.argo.org.cn). The ERA5 reanalysis are
available at the Copernicus Climate Change Service (C3S) Climate Date Store
(https://cds.climate.copernicus.eu/cdsapp#!/dataset/reanalysis-era5-single-levels). The along-track
Jason-3 satellite data are obtained from AVISO CNES Data Center (https://aviso-data-center.cnes.fr).
**Author contribution**
FX and RS designed the experiments and RS carried them out. RS developed the code of coupling
parametrizations and produced the figures. ZF contributed to the installation and operation of CFS2.0.
LL and HY contributed to the application of C-Coupler2. XL and YZ provided the original code of
CFS2.0. RS prepared the manuscript with contributions from all co-authors. FX contributed to review
and editing.
**Acknowledgments**
The authors would like to extend thanks to all developers of the CFS2.0 model
(https://cfs.ncep.noaa.gov/cfsv2/downloads.html). The work was supported by National Key Research
and Development Program of China (no. 2016YFC1401408), and Tsinghua University Initiative
Scientific Research Program (2019Z07L01001).





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
