# Peer review of "The Effects of Ocean Surface Waves on Global Forecast in CFS Modeling System v2.0"

_Geoscientific Model Development, 2020_

## Referee Comment (RC1) · Anonymous Referee #1 · 26 Nov 2020

This paper presents a study on the effects of surface gravity waves on several parameters obtained from a global simulation using the CFS2.0 climate model (including GFS at ∼100 km and MOM at 0.5°). The coupling effects between the wave model (WW3) and the climate model are the effect of the waves on the Langmuir mixing and on the wind stress. The tests are made on several 7-day time periods, either in summer or winter. They show that taking into account the Langmuir mixing reduces the warm SST bias and MLD shallow bias in the ACC, and that the impact of the waves on the roughness length reduces the positive bias of 10-m wind speed and SWH in mid latitude, even though this improvement is modest. This subject is interesting and taking into account the wave impact on simulating the ocean or the atmosphere at climate or NWP timescale has motivated several recent studies. The results and the method used here

are rather clearly presented. However, other aspects of the study are much less clear, namely I am not sure neither of the objectives of the study nor that they are addressed the right way. I list my major remarks below, I feel that both the interest and the impact of the study would be significantly improved should they be addressed.

1- I am not sure to understand the positioning of this study: the authors use a climate (OA coupled) model with a rather coarse horizontal resolution close to 100 km for the atmosphere and 0.5° for the ocean. These models are designed for long term studies at climate time scale, I am not aware of their use for NWP, which the authors present as the objective for their sensitivity study to coupling. This has some consequences on the processes that are sensitive to the coupling. Especially, I am not sure that the ocean (MLD and SST evolution) are so sensitive to the Langmuir mixing effect after a 7-day period but rather after months or years as is generally observed for oceanic processes in moderate conditions. If so, this should be assessed by comparing the results with those obtained using longer simulating time. Also, using such climate models for sensitivity tests at the NWP timescale should be justified and discussed. Are there any Met institutes currently using such models for weather forecasting? Global (coupled) models like IFS have horizontal resolutions close to 10 km and are able to represent explicitly mesoscale processes.

2- The methodology of the coupling with waves is insufficiently justified. About the wind-wave coupling several methods can be used, based either on the Charnock coefficient directly assessed in the wave model (ST4 in the paper) or parameterized from the wave age or wave steepness (ST4-Fan in the paper). Either has its merit, but the choice the authors made should be better discussed. Is it only based on the variation of z0 as shown Fig. 2? Also, in WW3, the ST3 parameterization is usually considered as the best suited for representing the high-frequency tail of the spectrum, which controls the Charnock coefficient, while ST4 is better suited for wave parameter modelling. So, I wonder why the ST3 parameterization is not considered here along with ST4. Did the authors test this? Please discuss. Same comments could be made about the wave-

ocean coupling. The authors consider only the effects on the Langmuir circulation and mixing (maybe on the Stokes-Coriolis force as well, this is not clear to me). Are other terms (see Couvelard et al. 2020 or Bao et al. 2020) not significant? Is this related to the timescale or the resolution considered?

3- Considering the resolution of the models and especially the short timescale of the experiment, I wonder whether the effects of the waves on the ocean are fully accounted for. Ocean has inertia, especially at 0.5° resolution, and I suspect that the effect of the coupling can increase or appear after the 7-day time period considered here, even with a hot start. Did the authors test longer simulations? I would like to see a comparison between the effects obtained after 7 days and a longer time period, to check that this choice is appropriate – and the corresponding discussion. Also, the diagram of coupling (Fig. 1) shows that surface currents are sent from the ocean model to WW3, but I suspect that their effect cannot be seen due to the coarse resolution of the ocean model (see Ardhuin et al. 2017, most of the current dynamics come from small scale).

4- The coupling with the atmosphere is not really a coupling but rather a change of parameterization, as well shown in Fig.2. As the authors correctly state in the text, the change in the wind speed or SWH comes from a mean change of z0 (mean difference between GFS and ST4 or ST4-Fan in Fig. 2), not from the variability induced by the waves (scattering of ST4 for instance). So changing the (constant) Charnock coefficient from 0.014 to e.g. 0.018 should have the same effect. This is a common confusion, as wave coupling is often mixed with change of parameterization (see Pianezze et al. 2018 for instance) but this be made clear in the text.

5- I am not sure about the significance of the results itself. Concerning the part I know best, the effect of the waves on the atmosphere, the decrease of the surface wind (and of the SWH) by an increase of the surface roughness, there is nothing really new here (see Renault et al. 2012, Pineau-Guillou et al. 2018 or Sauvage et al. 2020 for studies at NWP time scales). But the impact of waves can also change the heat fluxes and the atmospheric parameters through several mechanisms, which are discussed and

assessed in Renault et al. 2012 or Varlas et al. 2020. For instance, more mixing can change the SST and results in a decrease of the turbulent fluxes and air temperature. Could you please detail why you investigate only the effects on surface wind (among other atmospheric parameters)? The general improvement mentioned by the authors is rather modest (5%) and I wonder if considering other coupled effects or a longer time scale could change this. The statistical significance of the effects obtained is never discussed, for instance is the change of MLD between the different experiments shown in Fig. 5 significant with respect to the Argo values? Is the improvement of bias or scores for the SWH in Table 2 significant at 95 or 99%? Information about the number of data used for the comparison would be welcome, especially for along track satellite data. Also (minor point) the Percentage Absolute Difference rather looks like a relative score to me. Please clarify.

References

Ardhuin, F., S. T. Gille, D. Menemenlis, C. B. Rocha, N. Rascle, B. Chapron, J. Gula, and J. Molemaker (2017), Small-scale open ocean currents have large effects on wind wave heights, J. Geophys. Res. Oceans, 122, 4500– 4517, doi:10.1002/2016JC012413.

Bao, Y., Song, Z., & Qiao, F. (2020). FIO‐ESM version 2.0: Model description and evaluation. Journal of Geophysical Research: Oceans, 125, e2019JC016036. https://doi.org/ 10.1029/2019JC016036

Couvelard, X., Lemarié, F., Samson, G., Redelsperger, J. L., Ardhuin, F., Benshila, R., & Madec, G. (2020). Development of a two-way-coupled ocean–wave model: assessment on a global NEMO (v3. 6)–WW3 (v6. 02) coupled configuration. Geoscientific Model Development, 13(7), 3067-3090.

Pianezze, J., Barthe, C., Bielli, S.,Tulet, P., Jullien, S., Cambon, G., et al. (2018). A new coupled ocean-waves- atmosphere model designed for tropical storm studies: Example of tropical cyclone Bejisa (2013–2014) in the South-West

[Figure]

Indian Ocean. Journal of Advances in Modeling Earth Systems, 10, 801–825. https://doi.org/10.1002/2017MS001177

Pineau‐Guillou, L., Ardhuin, F., Bouin, M. N., Redelsperger, J. L., Chapron, B., Bidlot, J. R., & Quilfen, Y. (2018). Strong winds in a coupled wave–atmosphere model during a North Atlantic storm event: evaluation against observations. Quarterly Journal of the Royal Meteorological Society, 144(711), 317-332.

Renault, L., J. Chiggiato, J. C. Warner, M. Gomez, G. Vizoso, and J. TintoreÌĄ (2012), Coupled atmosphere-ocean-wave simulations of a storm event over the Gulf of Lion and Balearic Sea, J. Geophys. Res., 117, C09019, doi:10.1029/2012JC007924.

Sauvage, C., Lebeaupin Brossier, C., Bouin, M. N., & Ducrocq, V. (2020). Characterization of the air–sea exchange mechanisms during a Mediterranean heavy precipitation event using realistic sea state modelling. Atmospheric Chemistry & Physics, 20(3).

Varlas, G., Vervatis, V., Spyrou, C., Papadopoulou, E., Papadopoulos, A., & Katsafados, P. (2020). Investigating the impact of atmosphere–wave–ocean interactions on a Mediterranean tropical-like cyclone. Ocean Modelling, 153, 101675.

---

## Referee Comment (RC2) · Anonymous Referee #2 · 28 Nov 2020

This is another piece of work that is claiming that the coupling of an atmospheric model with the 3rd generation wave model for the specification of the surface momentum exchange is novel. Totally disregarding the work of Peter Janssen and colleagues at ECMWF. There is a whole book dedicated to the topic (Janssen 2004). The active two-way coupled system has been operational in ECMWF medium-range forecasting system since 1998, with frequent updates following thorough testing. See for instance the recent adaptation of ST4 based physics to the ECMWF IFS system (Bidlot 2019) and further enhancement of this parameterisation for tropical cyclone forecasts (Bidlot 2020). Robustness of the forecast performance requires many more cases. Obviously, this paper is a set of case studies. This needs to be clearly highlighted and discussed. In Janssen (2004), the impact of the coupling to waves is shown to be even more

important at longer lead time.

The addition of the coupling to an ocean circulation model in which a wave model interacts with both the atmosphere and the oceans for the purpose of medium-range forecast is not new either. Following the work of Breivik et al. (2015), all components of ECMWF forecast system have been fully coupled for the past few years. I agree that aspects of Upper Ocean mixing are still very crudely represented in many models. This study explores the potential of using the wave model surface Stokes drift to supplement a Langmuir mixing parameterisation. It is presented as a fast process acting quickly on the SST. But is it the right process? My understanding is that Langmuir turbulence might act much more slowly and is a factor in the determination of the mixed layer depth. Again, this study is a bit short to be really able to answer this question.

Is GFS only using the Charnock relation for the specification of the roughness length scale ($z0$) for momentum? There should also be a viscous contribution to $z0$. See Beljaars (1994). For this reason, I wonder about the coupling via $z0$. ST3 and ST4 in WW3 do not have a viscous term because they only deal with wave generation. From WW3, it is easy to determine the Charnock coefficient. Would it be more consistent to exchange it with GFS instead of $z0$? Moreover, the time steps of the different models imply that WW3 provides a new $z0$ every 900 s. Is it then kept constant until the next update? The Charnock relation implies that even if the Charnock coefficient is constant, the surface roughness can still change because the moment flux is still changing.

The implementation of ST4 in WW3 was selected. Noting that WW3 documentation suggests different set of values of the parameters for ST4, which one was used? The selections of the parameters for ST4 in WW3 were obtained by running experiments with the stand alone version of the code for given forcing in order to yield the best possible wave results, there is no absolute guarantee that the surface stress and hence $z0$ are what would be expected in a coupled system. There is obviously limited amount of observations of surface stress, however it would be reassuring that in the mean, the drag coefficient from WW3 is in agreement with field data as estimated from observations (Edson et al. 2013). It is also unclear to me that the surface stress that WW3 is specifying (via z0) is consistent with the surface stress used by MOM4, or is MOM4 surface stress specified using another bulk formula not necessarily consistent with the what prescribed by WW3. Finally, is the GFS formulation for heat and moisture fluxes dependent on z0 (as it is the case in the ECMWF system) and therefore the coupling with WW3 would also influence heat and moisture exchange (this can important in tropical cyclone simulation).

Some specific comments: In section 2.2, the wave model surface Stokes drift is used. The Stokes drift calculation from the wave model 2d spectrum is heavily weighted towards high frequency. Is the frequency cut-off in its calculation the same as the model cut-off ($\sim$0.41 Hz), without the addition of a high frequency tail? In this case, it would be probably be overly under estimated and one might wonder if a simpler parameterisation based on the wind speed will not suffice (ust(0) $\sim$ 0.016*U10), especially that it is mentioned that the potential misalignment between Ust and U10 has been found to be not important?

Anemometers mounted on buoys are rarely at 10m height. Nothing is mentioned regarding the adjustment of the buoy winds to 10m. The discussion regarding the bias reduction of 10m winds is only relevant if the buoy winds have been adjusted to 10m.

Minor comments: L50: you might want to add the following publications L63: modern reanalysis such as ERA5 is hourly L108: warm boots -> warm starts (?) Figure 1: so the low resolution surface currents are passed to the wave model, where the gradient in these is more important for wave refraction will therefore be poorly represented, but the same currents are not passed to the atmosphere where they could be used in a more consistent way to compute the momentum balance at the surface. L193: replace all reference to ERA5 by the Hersbach et al. (2020)

References:

Beljaars, A. C. M. 1994 : The parametrization of surface fluxes in large-scale models

under free convection. Q. J. R. Meteorol. Soc., 121, 255{270.

Jean-Raymond Bidlot, 2019: Model upgrade improves ocean wave forecasts. ECMWF newsletter, 159, 10-10. https://www.ecmwf.int/en/newsletter/159/news/model-upgrade-improves-ocean-wave-forecasts

Jean-Raymond Bidlot, Fernando Prates, Roberto Ribas, Anna Mueller-Quintino, Marijana Crepulja, Frédéric Vitart, 2020: Enhancing tropical cyclone wind forecasts, ECMWF newsletter, 164, 33-37. https://www.ecmwf.int/en/newsletter/164/meteorology/enhancing-tropical-cyclone-wind-forecasts

Edson, J.B., V. Jampana, R.A. Weller, S.P. Bigorre, A.J. Plueddemann, C.W. Fairall, S.D. Miller, L. Mahrt, D. Vickers and H. Hersbach, 2013: On the exchange of momentum over the open ocean. Journal of Physical Oceanography, 43:1589–1610, doi:10.1175/JPO-D-12-0173.1.

Hans Hersbach, Bill Bell, Paul Berrisford, Shoji Hirahara, Andras Horanyi, JoaquÄśnMunoz-Sabater, Julien Nicolas, Carole Peubey, Raluca Radu, Dinand Scheper1, AdrianSimmons, Cornel Soci, Saleh Abdalla, Xavier Abellan, Gianpaolo Balsamo, Peter Bechtold ,Gionata Biavati, Jean Bidlot, Massimo Bonavita, Giovanna De Chiara, Per Dahlgren, Dick Dee,Michail Diamantakis, Rossana Dragani, Johannes Flemming, Richard Forbes, Manuel Fuentes, Alan Geer, Leo Haimberger, Sean Healy, Robin J. Hogan, ElÄśas Holm, Marta Janiskova, Sarah Keeley, Patrick Laloyaux, Philippe Lopez, Cristina Lupu, Gabor Radnoti, Patricia de Rosnay, Iryna Rozum, Freja Vamborg, Sebastien Villaume, Jean-Noel Thepaut, 2020: The ERA5 Global Reanalysis, in final revision, Q. J. R. Meteorol. Soc. 00: 363 (2020).

Janssen, P , 2004: The Interaction of Ocean Waves and Wind. Cambridge, Cambridge University Press, doi:10.1017/CBO9780511525018.

Staneva J, Alari, V, Breivik O, Bidlot J.-R., Mogensen, K., 2016 : Effects of wave-

induced forcing on a circulation model of the North Sea. Ocean Dynamics (2016). doi:10.1007/s10236-016-1009-0

Wiese, A., Stanev, E., Koch, W., Behrens, A., Geyer, B., & Staneva, J. (2019). The Impact of the Two-Way Coupling between Wind Wave and Atmospheric Models on the Lower Atmosphere over the North Sea. Atmosphere, 10(7), 386.

―――――――――――――――――――

---

## Author Comment (AC1) · 24 Dec 2020

First, the authors would like to sincerely thank the reviewers for their careful reading of the paper and their valuable comments to the manuscript and helpful suggestions. We will modify the manuscript according to the comments in the next few weeks. In the following, our plans for revision of each comment are given.

Review from Referee #1

This paper presents a study on the effects of surface gravity waves on several parameters obtained from a global simulation using the CFS2.0 climate model (including GFS at ~100 km and MOM at 0.5°). The coupling effects between the wave model (WW3) and the climate model are the effect of the waves on the Langmuir mixing and on the

wind stress. The tests are made on several 7-day time periods, either in summer or winter. They show that taking into account the Langmuir mixing reduces the warm SST bias and MLD shallow bias in the ACC, and that the impact of the waves on the roughness length reduces the positive bias of 10-m wind speed and SWH in mid latitude, even though this improvement is modest. This subject is interesting and taking into account the wave impact on simulating the ocean or the atmosphere at climate or NWP timescale has motivated several recent studies. The results and the method used here are rather clearly presented. However, other aspects of the study are much less clear, namely I am not sure neither of the objectives of the study nor that they are addressed the right way. I list my major remarks below, I feel that both the interest and the impact of the study would be significantly improved should they be addressed.

1- I am not sure to understand the positioning of this study: the authors use a climate (OA coupled) model with a rather coarse horizontal resolution close to 100 km for the atmosphere and 0.5° for the ocean. These models are designed for long term studies at climate time scale, I am not aware of their use for NWP, which the authors present as the objective for their sensitivity study to coupling. This has some consequences on the processes that are sensitive to the coupling. Especially, I am not sure that the ocean (MLD and SST evolution) are so sensitive to the Langmuir mixing effect after a 7-day period but rather after months or years as is generally observed for oceanic processes in moderate conditions. If so, this should be assessed by comparing the results with those obtained using longer simulating time. Also, using such climate models for sensitivity tests at the NWP timescale should be justified and discussed. Are there any Met institutes currently using such models for weather forecasting? Global (coupled) models like IFS have horizontal resolutions close to 10 km and are able to represent explicitly mesoscale processes.

Response: As suggested, we will conduct longer simulations to test the effects of wave coupling and compare the effects of Langmuir mixing on the ocean (MLD and SST evolution) in 7-day forecast and longer periods. As stated by Saha et al. (2014), CFSv2

plays a role in the operational 6-10 day forecasts and two-week forecasts. CFSv2 seems to be ideal for sensitivity tests at NWP timescale and longer timescale. However, we have not found any Met institutes currently using such models for weather forecasting. Although global (coupled) models like IFS with higher resolution are able to represent mesoscale processes, the computational cost for the coupled global forecasting is too high. We will extend our forecasting period from 7-day to several months, and evaluate the sensitivity to coupling.

2- The methodology of the coupling with waves is insufficiently justified. About the wind-wave coupling several methods can be used, based either on the Charnock coefficient directly assessed in the wave model (ST4 in the paper) or parameterized from the wave age or wave steepness (ST4-Fan in the paper). Either has its merit, but the choice the authors made should be better discussed. Is it only based on the variation of z0 as shown Fig. 2? Also, in WW3, the ST3 parameterization is usually considered as the best suited for representing the high-frequency tail of the spectrum, which controls the Charnock coefficient, while ST4 is better suited for wave parameter modelling. So, I wonder why the ST3 parameterization is not considered here along with ST4. Did the authors test this? Please discuss. Same comments could be made about the wave-ocean coupling. The authors consider only the effects on the Langmuir circulation and mixing (maybe on the Stokes-Coriolis force as well, this is not clear to me). Are other terms (see Couvelard et al. 2020 or Bao et al. 2020) not significant? Is this related to the timescale or the resolution considered?

Response: As suggested, we will test the wave steepness-related parameterization and ST3 parameterization for the wind-wave coupling, and clearly discuss the merit of different parameterizations. About the wave-ocean coupling, we currently only test the effects of Langmuir mixing at the relative short time scale. We will add the Stokes-Coriolis force. The aim of this study is to evaluate the effects of different parameterizations of z0 and Stokes drift-related Langmuir mixing. To establish a more complete coupling system, more wave-related processes should be considered, such as sea

spray, wave breaking and non-breaking wave effects (Couvelard et al. 2020 or Bao et al. 2019). We will add these effects in our future study.

3- Considering the resolution of the models and especially the short timescale of the experiment, I wonder whether the effects of the waves on the ocean are fully accounted for. Ocean has inertia, especially at $0.5°$ resolution, and I suspect that the effect of the coupling can increase or appear after the 7-day time period considered here, even with a hot start. Did the authors test longer simulations? I would like to see a comparison between the effects obtained after 7 days and a longer time period, to check that this choice is appropriate – and the corresponding discussion. Also, the diagram of coupling (Fig. 1) shows that surface currents are sent from the ocean model to WW3, but I suspect that their effect cannot be seen due to the coarse resolution of the ocean model (see Ardhuin et al. 2017, most of the current dynamics come from small scale).

Response: As suggested, we will test a series of longer simulations. We compared tests with and without coupling of surface currents, the difference of the results is negligible. So, we will remove the pass of surface current to the wave model.

4- The coupling with the atmosphere is not really a coupling but rather a change of parameterization, as well shown in Fig.2. As the authors correctly state in the text, the change in the wind speed or SWH comes from a mean change of z0 (mean difference between GFS and ST4 or ST4-Fan in Fig. 2), not from the variability induced by the waves (scattering of ST4 for instance). So changing the (constant) Charnock coefficient from 0.014 to e.g. 0.018 should have the same effect. This is a common confusion, as wave coupling is often mixed with change of parameterization (see Pianezze et al. 2018 for instance) but this be made clear in the text.

Response: Although the change in the wind speed or SWH comes from the change of z0, we could not simply change the constant Charnock coefficient. As Shimura et al. (2017) indicated, a constant Charnock coefficient implies only wind speed-dependent roughness, while the change of z0 also depends on the development of waves. From

Fig.2, there is a spread of z0 at the same wind speed induced by wave variability. We will clarify this in the text.

5- I am not sure about the significance of the results itself. Concerning the part I know best, the effect of the waves on the atmosphere, the decrease of the surface wind (and of the SWH) by an increase of the surface roughness, there is nothing really new here (see Renault et al. 2012, Pineau-Guillou et al. 2018 or Sauvage et al. 2020 for studies at NWP time scales). But the impact of waves can also change the heat fluxes and the atmospheric parameters through several mechanisms, which are discussed and assessed in Renault et al. 2012 or Varlas et al. 2020. For instance, more mixing can change the SST and results in a decrease of the turbulent fluxes and air temperature. Could you please detail why you investigate only the effects on surface wind (among other atmospheric parameters)? The general improvement mentioned by the authors is rather modest (5%) and I wonder if considering other coupled effects or a longer time scale could change this. The statistical significance of the effects obtained is never discussed, for instance is the change of MLD between the different experiments shown in Fig. 5 significant with respect to the Argo values? Is the improvement of bias or scores for the SWH in Table 2 significant at 95 or 99%? Information about the number of data used for the comparison would be welcome, especially for along track satellite data. Also (minor point) the Percentage Absolute Difference rather looks like a relative score to me. Please clarify.

Response: As suggested, we will make the statistical significance test. After we extend the period of simulations from 7 days to several months, we will evaluate the change of heat fluxes and other atmospheric parameters such as 2-m air temperature, moisture and sea surface pressure. Previous studies mainly focused on the wave-related effects of global coupled systems in years and decades. While our evaluation focused on the effects of different parameterizations of z0 and Stokes drift-related Langmuir mixing in the CFSv2 in several days and several months, since CFSv2 is applicable in the relative short time scale. In addition, the Percentage Absolute Difference here is a

relative score, similar to the difference of mean absolute percentage error (MAPE) to evaluate the improve of bias.

References

Bao, Y., Song, Z., & Qiao, F.: FIO-ESM version 2.0: Model description and evaluation. Journal of Geophysical Research: Oceans, 125, e2019JC016036. https://doi.org/10.1029/2019JC016036, 2019

Couvelard, X., Lemarié, F., Samson, G., Redelsperger, J. L., Ardhuin, F., Benshila, R., & Madec, G.: Development of a two-way-coupled ocean–wave model: assessment on a global NEMO (v3. 6)–WW3 (v6. 02) coupled configuration. Geoscientific Model Development, 13(7), 3067-3090, 2020.

Saha, S., Moorthi, S., Wu, X., Wang, J., Nadiga, S., Tripp, P., Behringer, D., Hou, Y., Chuang, H., and Iredell, M. D.: The NCEP Climate Forecast System Version 2, Journal of Climate, 27, 2185-2208, http://dx.doi.org/10.1175/JCLI-D-12-00823.1, 2014.

Shimura T , Mori N , Takemi T , et al.: Long term impacts of ocean wave-dependent roughness on global climate systems. Journal of Geophysical Research: Oceans, 122(3), 2017.

---

## Author Comment (AC2) · 24 Dec 2020

First, the authors would like to sincerely thank the reviewers for their careful reading of the paper and their valuable comments to the manuscript and helpful suggestions. We will modify the manuscript according to the comments in the next few weeks. In the following, our plans for revision of each comment are given.

Review from Referee #2

This is another piece of work that is claiming that the coupling of an atmospheric model with the 3rd generation wave model for the specification of the surface momentum exchange is novel. Totally disregarding the work of Peter Janssen and colleagues at ECMWF. There is a whole book dedicated to the topic (Janssen 2004). The active

two-way coupled system has been operational in ECMWF medium-range forecasting system since 1998, with frequent updates following thorough testing. See for instance the recent adaptation of ST4 based physics to the ECMWF IFS system (Bidlot 2019) and further enhancement of this parameterization for tropical cyclone forecasts (Bidlot 2020). Robustness of the forecast performance requires many more cases. Obviously, this paper is a set of case studies. This needs to be clearly highlighted and discussed. In Janssen (2004), the impact of the coupling to waves is shown to be even more important at longer lead time.

Response: We acknowledge the pioneering work in ECMWF. We sincerely apologized for any misleading statement in the study. As indicated by the referee, our study focused on the effects of different parameterizations of z0 and Stokes drift-related Langmuir mixing in the CFSv2 in a series of cases. We will make it clear in the revised manuscript. Though Janssen (2004) suggested the impact of the coupling to waves is shown to be even more important at longer lead time, it is still interesting to investigate the effects of waves in a relative short period. We will extend the period of simulations from 7 days to several months, and evaluate more variables, such as 2-m air temperature, moisture and sea surface pressure etc. to investigate the effects of wave coupling in CFSv2.

The addition of the coupling to an ocean circulation model in which a wave model interacts with both the atmosphere and the oceans for the purpose of medium-range forecast is not new either. Following the work of Breivik et al. (2015), all components of ECMWF forecast system have been fully coupled for the past few years. I agree that aspects of Upper Ocean mixing are still very crudely represented in many models. This study explores the potential of using the wave model surface Stokes drift to supplement a Langmuir mixing parameterization. It is presented as a fast process acting quickly on the SST. But is it the right process? My understanding is that Langmuir turbulence might act much more slowly and is a factor in the determination of the mixed layer depth. Again, this study is a bit short to be really able to answer this question.

Response: The Langmuir circulation is also important at weather time scale. As Kukulka et al. (2009) indicated, the modification of Langmuir circulation to upper temperature profiles could be produced quickly in a few hours. We agree that the time period in this study is too short, so we will make longer period simulations to answer the question.

Is GFS only using the Charnock relation for the specification of the roughness length scale (z0) for momentum? There should also be a viscous contribution to z0. See Beljaars (1994). For this reason, I wonder about the coupling via z0. ST3 and ST4 in WW3 do not have a viscous term because they only deal with wave generation. From WW3, it is easy to determine the Charnock coefficient. Would it be more consistent to exchange it with GFS instead of z0? Moreover, the time steps of the different models imply that WW3 provides a new z0 every 900 s. Is it then kept constant until the next update? The Charnock relation implies that even if the Charnock coefficient is constant, the surface roughness can still change because the moment flux is still changing.

Response: We agree that there should be a viscous contribution to z0 especially when the sea surface is smooth and wind speeds are low. As suggested, we will exchange the Charnock coefficient with GFS instead of z0. We will also modify the time step of WW3 to be the same 180 s as CFS in all simulations.

The implementation of ST4 in WW3 was selected. Noting that WW3 documentation suggests different set of values of the parameters for ST4, which one was used? The selections of the parameters for ST4 in WW3 were obtained by running experiments with the stand alone version of the code for given forcing in order to yield the best possible wave results, there is no absolute guarantee that the surface stress and hence z0 are what would be expected in a coupled system. There is obviously limited amount of observations of surface stress, however it would be reassuring that in the mean, the drag coefficient from WW3 is in agreement with field data as estimated from observations (Edson et al. 2013). It is also unclear to me that the surface stress that WW3 is specifying (via z0) is consistent with the surface stress used by MOM4, or is MOM4

surface stress specified using another bulk formula not necessarily consistent with the what prescribed by WW3. Finally, is the GFS formulation for heat and moisture fluxes dependent on z0 (as it is the case in the ECMWF system) and therefore the coupling with WW3 would also influence heat and moisture exchange (this can important in tropical cyclone simulation).

Response: The parameters we used here are the TEST471, which is commonly-used at global scale. But we will further compare its effect with that of TEST471f, since the TEST471f is the CFSR tuned ST4 setup and might be more suitable in our system. We agree that the selections of different input-dissipation source term (ST) need to be further discussed in a coupled system. We will conduct a series of simulations to test the effect of ST3 (BJA) parameterization, whose performance is almost as good as ST4 in a single WW3. And as suggested, we will compare the simulated drag coefficient with observations (Edson et al. 2013). About the surface stress (via z0) in MOM4, it is the same as that in GFS due to coupling, and hence it is consistent with what prescribed by WW3. Yes, the GFS formulation for heat and moisture fluxes is also dependent on z0. Therefore the coupling with WW3 would also influence heat and moisture exchange.

Some specific comments: In section 2.2, the wave model surface Stokes drift is used. The Stokes drift calculation from the wave model 2d spectrum is heavily weighted towards high frequency. Is the frequency cut-off in its calculation the same as the model cut-off ($\sim$0.41 Hz), without the addition of a high frequency tail? In this case, it would be probably be overly under estimated and one might wonder if a simpler parameterization based on the wind speed will not suffice (ust(0)$\sim$0.016*U10), especially that it is mentioned that the potential misalignment between Ust and U10 has been found to be not important?

Response: Yes, the cut-off frequency of Stokes drift is the same as the model cut-off. We will increase the cut-off frequency and test the effects. According to Li et al. (2016), they included high frequency tail assumption for Stokes drift, but their results

show the effect of misalignment between ust and U10 are rather small under a relatively coarse spatial resolution. In addition, they tested the ust bias with increasing spatial resolutions in a rather wide frequency range, and the bias differences are negligible. Thus, although their spatial resolutions are coarser than ours, we believe in our system even with the high frequency tail assumption the misalignment between ust and U10 may still not be clearly seen. These need further exploration.

Anemometers mounted on buoys are rarely at 10m height. Nothing is mentioned regarding the adjustment of the buoy winds to 10m. The discussion regarding the bias reduction of 10m winds is only relevant if the buoy winds have been adjusted to 10m.

Response: We will use the method (Hsu et al. 1994) recommended by NDBC to adjust the buoy winds to 10 m.

Minor comments: L50: you might want to add the following publications L63: modern reanalysis such as ERA5 is hourly L108: warm boots -> warm starts (?) Figure 1: so the low resolution surface currents are passed to the wave model, where the gradient in these is more important for wave refraction will therefore be poorly represented, but the same currents are not passed to the atmosphere where they could be used in a more consistent way to compute the momentum balance at the surface. L193: replace all reference to ERA5 by the Hersbach et al. (2020).

Response: We will modify the manuscript according to these comments. Since the gradients of surface currents are poorly represented in the model, we will remove the pass of surface currents to the wave model.

References

Edson, J.B., V. Jampana, R.A. Weller, S.P. Bigorre, A.J. Plueddemann, C.W. Fairall, S.D. Miller, L. Mahrt, D. Vickers and H. Hersbach: On the exchange of momentum over the open ocean. Journal of Physical Oceanography, 43:1589–1610, doi:10.1175/JPO-D-12-0173.1, 2013.

Hsu, S. A., Eric A. Meindl, and David B. Gilhousen, 1994: Determining the Power-Law Wind-Profile Exponent under Near-Neutral Stability Conditions at Sea, Applied Meteorology, Vol. 33, No. 6, 1994.

Janssen, P., and Janssen, P. A.: The interaction of ocean waves and wind, Cambridge University Press, 2004, pp 312.

Kukulka, T., Plueddemann, A. J., Trowbridge, J. H., and Sullivan, P. P.: Significance of Langmuir circulation in upper ocean mixing: comparison of observations and simulations, Geophysical Research Letters, 36, http://dx.doi.org/10.1029/2009GL037620, 2009.

---

## Author Comment (AC3) · 28 Dec 2020

We have a little more to add to the significance of our research. The CFS is a coupled system with wide application and great development potential. The CFS Reanalysis (CFSR) data is widely used in other models, such as providing the initial field for WRF and as the forcing field for WW3. In addition, the NCEP is establishing its own first global scale coupled system, called GFS-Wave (https://polar.ncep.noaa.gov/waves/wavewatch), in which the Global Forecast System (GFS; the atmosphere module in CFS system) is one-way coupled with the WW3 global wave model. Our work provides test cases of two-way wave coupling of CFS on weather and sub-seasonal forecast. The impact of two-way wave coupling is clearly shown. The study would be useful for the future development of CFS-wave coupling

system.

---

## Author Comment (AC4) · 4 Jan 2021

According to comments, we have improved the ocean mixing process related to the Stokes drift by adding Stokes-Coriolis force and entrainment. The preliminary results of SST from a series of 14-day forecasts were shown in Fig. 1&2 and the legends were consistent with Fig. 3 in the article. Compared with the original VR12-AL-ONLY experiment (Fig. 1c&2c), both Stokes-Coriolis force (Fig. 1d&2d) and entrainment (Fig. 1e&2e) can further improve the SST biases, and the combination effects (Fig. 1f&2f) were more obvious. Compared with the results on the 7th day (Fig. 1), on the 14th day (Fig. 2) the improvement was strengthened (blue area), but the increase of bias caused by excessive mixing was also strengthened (red area). Next, we will further discuss the z0-realted parameterizations and conduct simulations with a longer time.